# MAPK Inhibition Requires Active RAC1 Signaling to Effectively Improve Iodide Uptake by Thyroid Follicular Cells

**DOI:** 10.3390/cancers13225861

**Published:** 2021-11-22

**Authors:** Márcia Faria, Rita Domingues, Maria João Bugalho, Paulo Matos, Ana Luísa Silva

**Affiliations:** 1Serviço de Endocrinologia, Diabetes e Metabolismo, do Centro Hospitalar Universitário de Lisboa Norte-Hospital Santa Maria, 1649-035 Lisboa, Portugal; mcsf.faria@gmail.com (M.F.); rita.sofia.domingues@gmail.com (R.D.); maria.bugalho@chln.min-saude.pt (M.J.B.); 2BioISI-Biosystems and Integrative Sciences Institute, Faculdade de Ciências da Universidade de Lisboa, 1749-016 Lisboa, Portugal; paulo.matos@insa.min-saude.pt; 3Departamento de Genética Humana, Instituto Nacional de Saúde Doutor Ricardo Jorge, 1649-016 Lisboa, Portugal; 4ISAMB-Instituto de Saúde Ambiental, Faculty of Medicine, University of Lisbon, 1649-035 Lisboa, Portugal; 5Faculty of Medicine, University of Lisbon, 1649-035 Lisboa, Portugal

**Keywords:** NIS, RAC1, thyroid cancer, radioiodide uptake, RAI therapy, MAPK inhibitors

## Abstract

**Simple Summary:**

The Sodium/Iodide Simulator (NIS) is responsible for the uptake of iodide in the thyroid follicular cells. NIS is present in most differentiated thyroid carcinomas (DTC), allowing radioactive iodine (RAI) to be used to destroy malignant cells. However, a significant proportion of DTCs stop picking up iodide and become resistant to RAI therapy. This is mainly due to the symporter no longer being produced or not being placed correctly at the cell’s membrane. This has been associated with mechanisms linked to malignant transformation, namely the overactivation of the so-called MAPK pathway. Thus, several drugs have been developed to inhibit this pathway, attempting to increase NIS levels and iodide uptake. However, MAPK inhibitors have had only partial success in restoring NIS expression. We found that the activity of another protein, the small GTPase RAC1, has an important role in this process, determining the outcome of MAPK inhibitors. Thus, our findings open new opportunities to find effective therapeutic alternatives for DTC resistant to RAI.

**Abstract:**

The Sodium/Iodide Symporter (NIS) is responsible for the active transport of iodide into thyroid follicular cells. Differentiated thyroid carcinomas (DTCs) usually preserve the functional expression of NIS, allowing the use of radioactive iodine (RAI) as the treatment of choice for metastatic disease. However, a significant proportion of patients with advanced forms of TC become refractory to RAI therapy and no effective therapeutic alternatives are available. Impaired iodide uptake is mainly caused by the defective functional expression of NIS, and this has been associated with several pathways linked to malignant transformation. MAPK signaling has emerged as one of the main pathways implicated in thyroid tumorigenesis, and its overactivation has been associated with the downregulation of NIS expression. Thus, several strategies have been developed to target the MAPK pathway attempting to increase iodide uptake in refractory DTC. However, MAPK inhibitors have had only partial success in restoring NIS expression and, in most cases, it remained insufficient to allow effective treatment with RAI. In a previous work, we have shown that the activity of the small GTPase RAC1 has a positive impact on TSH-induced NIS expression and iodide uptake in thyroid cells. RAC1 is a downstream effector of NRAS, but not of BRAF. Therefore, we hypothesized that the positive regulation induced by RAC1 on NIS could be a relevant signaling cue in the mechanism underlying the differential response to MEK inhibitors, observed between NRAS- and BRAF-mutant tumors. In the present study, we found that the recovery of NIS expression induced through MAPK pathway inhibition can be enhanced by potentiating RAC1 activity in thyroid cell systems. The negative impact on NIS expression induced by the MAPK-activating alterations, NRAS Q61R and BRAF V600E, was partially reversed by the presence of the MEK 1/2 inhibitors AZD6244 and CH5126766. Notably, the inhibition of RAC1 signaling partially blocked the positive impact of MEK inhibition on NIS expression in NRAS Q61R cells. Conversely, the presence of active RAC1 considerably improved the rescue of NIS expression in BRAF V600E thyroid cells treated with MEK inhibitors. Overall, our data support an important role for RAC1 signaling in enhancing MAPK inhibition in the context of RAI therapy in DTC, opening new opportunities for therapeutic intervention.

## 1. Introduction

The active transport of iodide into thyroid follicles is mediated by the sodium/iodide symporter (NIS), an integral plasma membrane glycoprotein protein present at the basolateral surface of thyroid follicular cells. It belongs to the human solute carrier (SLC) family of transporters, and uptakes iodide through the co-transport of 2Na^+^ and one I^−^ from the bloodstream, using the Na^+^ gradient maintained by Na^+^/K^+^ ATPase. NIS is highly expressed in the thyroid and plays an essential role in thyroid hormone synthesis, of which iodine is an essential constituent [1].

The functional expression of NIS is usually retained in well-differentiated thyroid carcinomas (DTCs). This allows the use of radioactive iodine (RAI) as a diagnostic and therapeutic tool. In fact, the selective use of RAI, based on patients’ individual risks, is often recommended as an adjuvant treatment for DTC, for which the standard care includes total thyroidectomy and thyroid-stimulating hormone (TSH)-suppressive therapy. In the presence of metastatic disease, RAI is the treatment of choice. Notwithstanding, a significant proportion of patients with advanced metastatic disease fail to respond to RAI therapy (RAI-refractory TC). For these patients, no effective therapeutic alternatives are available, which drastically reduces their survival rate [2].

TSH is the central regulator of NIS expression in the thyroid gland. TSH stimulation is a primary requirement for the full activation of NIS expression and is also necessary for the optimal localization of NIS at the plasma membrane [3,4].

Despite NIS expression being retained in most DTCs, the levels of NIS and iodide uptake are reduced when compared to normal thyroid tissue [5]. A defective NIS functional expression is the main reason for impaired iodide uptake and has been associated with several pathways linked to malignant transformation [6]. In fact, the canonical RAS/RAF/MEK/ERK pathway (mitogen-activated protein kinase (MAPK) pathway), which has emerged as a key signaling pathway involved in thyroid tumorigenesis [7,8], has been associated with impaired NIS expression and function [9]. BRAF [murine sarcoma viral (v-raf) oncogene homolog B1] and RAS (rat sarcoma virus homolog) gain-of-function mutations are two genetic events closely associated with DTCs that constitutively activate the MAPK pathway [10]. The BRAF V600E mutation is the most frequent genetic alteration in the papillary DTC subtype (PTC) and is widely described in association with the impaired expression of thyroid-specific genes, such as NIS. Accordingly, decreased NIS expression at both the transcript and protein levels [11,12,13,14], as well as impaired targeting of the NIS to the basolateral membrane, has been demonstrated in BRAF-mutant tumors [14,15,16,17,18,19,20]. RAS-activating mutations are less common in PTC, although are the most prevalent genetic alterations in the differentiated follicular thyroid cancer (FTC). Despite also being associated with decreased NIS expression, RAS-activating mutations have a reduced impact on the expression of the symporter compared to BRAF V600E [15].

Several strategies focused on the inhibitors of the MAPK pathway have been developed with the goal of increasing the uptake of iodide in refractory-thyroid cancer (TC) [21,22,23,24,25,26]. A study by Chakravarty and collaborators demonstrated that BRAF and MEK inhibitors were able to restore RAI uptake in mouse thyroid carcinomas turned refractory to iodide by the conditional activation of BRAF [27]. In humans, the relevance of this strategy was documented in a clinical trial by Ho and collaborators [25] that observed a meaningful gain in RAI uptake in a subgroup of patients with refractory-TC, after treatment with the MEK1/2 inhibitor, AZD6244 (Selumetinib). Unexpectedly, the therapeutic benefit was more prominent in NRAS-mutant tumors (five out of five showed iodide resensitization) compared to BRAF-mutant tumors, where only one out of nine patients showed increased RAI uptake [25]. The cause of this disparity remains elusive.

In a previous study, we showed that NIS expression can be enhanced by the activity of RAC1, a member of the RAS superfamily of small GTPases. These comprise a class of molecular “switches” that regulate signaling pathways involved in processes such as gene expression, cell proliferation and cell migration [28].

RAC1 is a downstream effector of NRAS, but not of BRAF [29,30,31]. Therefore, we hypothesized that the positive regulation induced by RAC1 on NIS could be a relevant signaling cue in the mechanism underlying the differential response to the MEK inhibitor AZD6244, observed between NRAS- and BRAF-mutant tumors. In this way, the RAI resensitization induced by MAPK inhibition in NRAS-mutant tumors would be potentiated by the NRAS-mediated induction of RAC1, a positive regulator of NIS.

## 2. Materials and Methods

### 2.1. Cell Lines and Culture Conditions

In this study, we used the cell lines PCCL3 (BCRJ 0204) and FRTL5 (ECACC91030711), both derived from Fischer rats’ normal thyroid follicular epithelium responsive to TSH stimulation. The Y-PCCL3 cells that stably express the halide-sensitive yellow fluorescent protein HS-YFP-F46L/H148Q/I152L cells were generated from parental PCCL3, as previously described [32]. Both cell lines were cultured in F-12 Coon’s modified liquid medium (Merck, Darmstadt, Germany) supplemented with Fetal Bovine Serum (5%, FBS, Gibco, Grand Island, NY, USA), insulin (10 μg/mL)), Apo-Transferrin (5 μg/mL, Apo-T) and TSH (0.1 mU/mL) (all from Sigma-Aldrich, St. Louis, MO, USA). A starvation medium (F12 Coon’s modified medium supplemented with FBS (0.2% *v/v*) and Apo-T (5 μg/mL) or stimulation medium (F12 Coon’s modified medium supplemented with FBS (5% *v/v*), Apo-T (5 μg/mL) and TSH (1 mU/mL) were used when indicated.

The human PTC-derived TPC-1 (CVCL_6298) and BCPAP (CVCL_0153) cell lines were maintained in RPMI (Gibco, Grand Island, NY, USA), both supplemented with 10% *v/v* FBS. All cells were maintained at 37 °C in a 5% humidified CO_2_ environment, regularly checked for the absence of mycoplasm infection and discarded after 20 passages.

Treatment of cells with the RAC1 selective inhibitor NSC23766 (100 µM, Santa Cruz Biotechnology, Santa Cruz, CA, USA) was performed for the indicated time periods in the appropriate supplemented medium. Treatment of cells with the MEK-inhibitors AZD6244 (10 µM, APExBio, Oss, The Netherlands) or CH5126766 (10 µM, APExBio, Oss, The Netherlands) was performed for 48 h in the appropriate supplemented medium.

### 2.2. Plasmids and Transfections

The TPC-1 cell line was used for the stable ectopic expression of HA-tagged NIS [pECFP-N1 plasmid- (BD Biosciences Clontech, San Carlos, CA, USA) derived construct expressing the SLC5A5 gene coding sequence (Ref Seq NM_000453.2) with an in-frame insertion coding a triple hemagglutinin tag (HA) in the proteins’ 9th extracellular loop, and CFP exclusion]. Positive HA-NIS cell clones were selected with 1000 μg/mL Geneticin (Gibco, Grand Island, NY, USA) and further validated (HA-NIS cell surface residency was confirmed by surface protein biotinylation assay and NIS ability to uptake iodide was confirmed by HS-YFP-based iodide influx [33]). BCPAP cells were transfected with pEGFP-RAC1-L61 plasmid as previously described [32]. The pEGFP-C3 (BD Biosciences Clontech, San Carlos, CA, USA) was used for mock controls. All transfections were performed with Lipofectamine 2000, according to the manufacturer’s instructions, using a 1:4 (μg/μL) DNA: reagent ratio.

### 2.3. Lentiviral Production and Cell Transductions

pCDH-CFP, pCDH-CFP-NRAS Q61R and pCDH-CFP-BRAF V600E lentiviral expression vectors were packaged into lentiviral particles using the psPAX2 packaging plasmid DNA (Addgene #12260) and the pMD2.G envelope plasmid DNA (Addgene #12259). Briefly, 2.5 × 10^6^ HEK293T (human embryonic kidney; CVCL_0063) cells were plated onto poly-L-lysine- (Sigma-Aldrich) coated 25 cm^2^ flasks. Twenty-four hours later, cells at 70% confluence were co-transfected with 2.5 µg of each pCDH-vector, 1.5 µg psPAX2 and 1 µg pMD2.G, using METAFECTENE (Biontex, München, Germany) at a final ratio of 1 µg DNA: 4.5 µL METAFECTENE. The medium was removed 6 h later and replaced with 5 mL of fresh DMEM supplemented with 10% FBS. CFP fluorescence was monitored to confirm transfection efficiency. Culture supernatants were collected after 72 h and cell debris were pelleted by centrifugation at 300× *g* in an Eppendorf 5810 centrifuge. Lentiviral supernatants were then filtered through a 0.45 μM filter and stored for further usage in cell transduction. Briefly, PCCL3 or FRTL5 cells were grown in 6-well plates to 60% confluence and incubated with 1 mL lentiviral supernatants and 8 µg/mL polybrene (Sigma-Aldrich, St. Louis, MO, USA). 6-well plates were spun for 30 min at 37 °C at 1200× *g*. 48 h after transduction, cells were treated accordingly to each experimental assay.

### 2.4. RNA Extraction and RT-qPCR

Total RNA extraction from cultured cells was performed using the ready-to-use reagent TripleXtractor (GRiSP Research Solutions, Porto, Portugal). Two micrograms of RNA were used for cDNA synthesis using RevertAid Reverse Transcriptase (Thermo Scientific, Waltham, MA, USA) and random primers (Roche, Mannheim, Germany). All these standardized procedures were performed according to the manufacturer’s instructions. The SYBR Green- (Nzytech, Lisboa, Portugal) based qPCR experimental procedure previously described [32] was used to quantify NIS expression levels, which were normalized to the endogenous control expression levels of the HPRT1 and TBP genes of rats and humans, when respectively indicated.

### 2.5. CRIB Pull-Down Assays for RAC1 Activation Status

CRIB pull-down assays were performed as previously described [34]. Briefly, lysates from PCCL3, FRTL5 or TPC1 cells, transduced or treated as indicated, were incubated with the CRIB (CDC42/RAC1 Interactive Binding) peptide domain of PAK1 (that specifically binds to the GTP-bound form of RAC1 protein) pre-coupled to streptavidin-agarose beads (Sigma-Aldrich, St. Louis, MO, USA). Input and pulled-down protein fractions, solubilized in 2 × Laemmli buffer [34], were then analyzed by western blot.

### 2.6. Cell Surface Protein Biotinylation Assay

The cell surface protein biotinylation assay was performed as described [33]. The arrest of endocytic traffic was attained by incubating cells with ice-cold PBS–CM (PBS pH 8.0 containing 1 mM CaCl_2_ and 1 mM MgCl_2_) for 5 min. Cell surface proteins were then labeled for 45 min on ice with sulfosuccinimidyl 3-[[2-(Biotinamido)ethyl] di-thio] propionate sodium salt (0.5 mg/mL, sc-212981, Santa Cruz Biotechnology, Santa Cruz, CA, USA). Non-reacting biotin molecules were inactivated with quenching buffer (100 mM Tris/HCl pH 7.5, 150 mM NaCl, 1 mM CaCl_2_, 1 mM MgCl_2_, 10 mM glycine, 1% (*w/v*) BSA), and cells were lysed in pull-down buffer [50 mM Tris/HCl pH 7.5, 100 mM NaCl, 10% (*v/v*) glycerol, 1% (*v/v*) NP40, 0,1% (*v/v*) SDS supplemented with protease inhibitor cocktail composed of 1 mM PMSF, 1 mM 1,10-phenanthroline, 1 mM EGTA, 10 μM E64, and 10 μg/mL of each aprotinin, leupeptin and pepstatin A (all from Sigma-Aldrich, St. Louis, MO, USA)]. An aliquot of each cell lysate, representing input protein levels, was collected and the remaining lysate was incubated with streptavidin-conjugated agarose beads (Sigma-Aldrich, St. Louis, MO, USA). Captured proteins were solubilized 2 × in Laemmli buffer supplemented with 100 mM dithiothreitol (DTT)] and analyzed by western blotting.

### 2.7. Western Blot

Western blots were performed using standard protocols: protein lysates were resolved in 10% SDS-PAGE and transferred to PVDF membranes (Bio-Rad, Hercules, CA, USA). The primary antibodies and dilutions used were the following: rabbit polyclonal anti-NIS (1:500; 24324-1-AP, Proteintech, Rosemont, IL, USA); rabbit polyclonal anti-GFP (1:10,000; ab290, Abcam, Cambridge, UK); rabbit poly-clonal anti-ERK 1/2 antibody (1:1000; #4695, Cell Signaling, Danvers, MA, USA); mouse monoclonal anti-HA (1:10,000; 11 583 816 001, Roche, Mannheim, Germany); mouse monoclonal anti-phospho-ERK anti-body (1:1000; M8159, Sigma-Aldrich, St. Louis, MO, USA); mouse monoclonal anti-PCNA (1:2000; NA03, Merck, Darmstadt, Germany); mouse monoclonal anti-RAC1 (1:1000; 05-389, Millipore, Burlington, MA, USA) and mouse monoclonal anti-alpha tubulin (1:300,000; T5168, Sigma-Aldrich, St. Louis, MO, USA). HRP-labeled anti-rabbit or anti-mouse IgG secondary antibodies (1:3000; Bio-Rad, Hercules, CA, USA) were used for chemiluminescence imaging.

### 2.8. HS-YFP–Based Iodide Influx Assay

Iodide (I^−^) influx was accessed as previously described [32]. Y-PCCL3, transduced as indicated, were serum-starved for 24 h and subsequently stimulated for 96 h with 1 mU/mL TSH. When appropriate, cells were also treated with the RAC1 inhibitor NSC23766 or the MEK-inhibitor AZD6244, as indicated. After the addition of isomolar PBS–NaI solution (1 mM final NaI concentration), the decay of YFP fluorescence was followed for 500 s by recording the fluorescence by acquiring an image every 10 s in a Leica fluorescence microscope. NIS-mediated iodide uptake specificity was attested with ClO4- (1 mM), a competitive inhibitor of NIS. Cell fluorescence recordings were obtained through analysis with Image J (NIH, Bethesda, MD, USA) and normalized for the initial average value measured before the addition of iodide. Initial iodide influx rates were calculated by fitting the fluorescence decay curves to exponential functions as previously described [35].

### 2.9. Statistical Analysis

Statistical analysis was performed using GraphPad Prism 5 software (GraphPad Software, San Diego, CA, USA). Quantitative results are shown as the means ± the standard error of the mean (SEM) from three to five independent experiments, as indicated. One-way ANOVA followed by post-hoc Tukey HSD or Dunnett’s tests were used for multiple comparisons and two-tailed, unpaired student’s *t*-tests were applied to compare two sets of data. In all analyses the significance level was set at 0.05 and *p*-values were represented as follows: *p* < 0.05 (*), *p* < 0.01 (**) and *p* < 0.001 (***).

## 3. Results

### 3.1. Both NRAS Q61R and BRAF V600E Decrease NIS Expression, but Only NRAS Q61R Promotes RAC1 Activation

To ascertain the impact of NRAS- and BRAF- derived signaling on NIS expression, we started this study by comparing the impact of NRAS Q61R and BRAF V600E mutations on NIS endogenous expression in a thyroid cell model representative of a TSH-responsive system for NIS expression and iodide uptake. For this, we used the non-neoplastic rat thyroid cell line PCCL3, in which the overexpression of the constitutively active NRAS Q61R and BRAF V600E mutants was induced using a lentiviral expression system. Transduced cells were stimulated with TSH and the effect on NIS mRNA levels was accessed by RT-qPCR. The overexpression of both NRAS Q61R and BRAF V600E mutants induced a significant decrease in NIS transcript levels upon TSH stimulation (Figure 1A). This same effect was observed in FRTL5 cells, another normal rat thyroid cell line (Appendix A), supporting that both genetic alterations have a similar impact on NIS expression.

RAC1 was previously pointed out as a key downstream effector of NRAS Q61K for tumor growth in melanoma [30], but no study has been conducted in thyroid cancer. This prompted us to analyze the effect of NRAS and BRAF mutants on RAC1 activation statuses in thyroid cell models. Using a CRIB pull-down assay, which allows selective capture of the active, GTP-bound pool of RAC1 in cells [34], we observed that NRAS Q61R but not BRAF V600E overexpression increased the levels of intracellular-active RAC1 in both PCCL3 (Figure 1B) and FRTL5 cells (Appendix A). These results are consistent with RAC1 being selectively activated downstream of NRAS but not of BRAF in thyroid cells.

### 3.2. MEK Inhibition Reverts the Negative Effect of NRAS Q61R on NIS Transcript Levels but Not That of BRAF V600E

Next, we compared the ability of the MEK inhibitor AZD6244 to increase NIS mRNA levels in the context of MAPK pathway overactivation by either NRAS or BRAF gain-of-function mutations. As expected, in the presence of TSH, AZD6244 treatment increased NIS expression in NRAS Q61R and BRAF V600E-expressing cells, but not in control PCCL3 cells (Figure 2). A similar impact of AZD6244 on NIS expression was observed for FRLT5 cells also expressing the activating mutants of NRAS and BRAF (Appendix A). Interestingly, AZD6244-mediated induction of NIS was considerably more effective in NRAS Q61R-expressing cells than in cells harboring BRAF V600E. In fact, AZD6244 treatment restored NIS expression nearly to control levels in NRAS-mutant cells but was only partially effective in cells expressing mutant BRAF (Figure 2 and Appendix A). This is consistent with the differential response to AZD6244 previously reported in PTC patients [25].

Notably, in certain cellular contexts, particularly when BRAF V600E-activating mutations are present, the inhibitory effect of AZD6244 on the MAPK pathway can be transient [36]. In contrast, CH5126766, another allosteric MEK inhibitor, has been shown to induce a sustained inhibition of this pathway by also inhibiting the BRAF kinase [36]. Hence, we asked whether CH5126766 would have a different impact on NIS expression in thyroid cells bearing NRAS Q61R versus BRAF V600E backgrounds.

Interestingly, similar to AZD6244, the recovery of NIS expression induced by CH5126766 was only partial in cells expressing BRAF V600E, while bringing NIS transcript levels close to those of control in NRAS-mutant cells, in both cell models tested (Figure 2 and Appendix A). Phospho-ERK protein levels were used as a readout for MAPK pathway activity, confirming its upregulation in NRAS- and BRAF-mutant cells (Figure 2B and Appendix A). Treatments with either AZD6244 or CH5126766 efficiently inhibited the MAPK pathway in both mutant and control cells, inducing a large decrease in ERK phosphorylation (Figure 2B and Appendix A).

### 3.3. Inhibition of RAC1 Signaling Hampers the Rescue of NIS Expression Mediated by MEK Inhibitors in NRAS Q61R Cells

Given the differential recovery of NIS expression in response to MEK inhibition observed between NRAS- and BRAF-activated signaling, we asked whether the activation of RAC1 would be involved in this effect. In fact, we have previously demonstrated that RAC1 has a positive impact on TSH-induced NIS expression in both PCCL3 and FRTL5 cells [32], and we showed above, that RAC1 activation is increased by NRAS Q61R ectopic expression but not by BRAF V600E, in both cell models tested.

To further investigate this hypothesis, TSH-stimulated NRAS Q61R- or BRAF V600E-expressing PCCL3 cells were treated with AZD6244 or co-treated with AZD6244 and the RAC1-selective inhibitor NSC23766. The efficacy of NSC23766 treatment was confirmed by the significant decrease in the active pool of endogenous RAC1 molecules upon exposure to this inhibitor in control, BRAF V600E- and, even more substantially, in NRAS Q61R-expressing cells (Figure 3A). NIS mRNA levels in these conditions were assessed by RT-qPCR, as above. The negative impact of chemical RAC1 inhibition on NIS expression was apparent in parental cells (Figure 3B), as shown previously [32]. Notably, the NIS expression induced by AZD6244in cells expressing NRAS Q61R was abolished by RAC1 inhibition but the RAC1 inhibitor did not significantly affect AZD6244-induced NIS expression in BRAF V600E-expressing cells (Figure 3B). Similar results were obtained when using FRTL5 cells (Appendix A).

Lastly, we confirmed that the observed effects in NIS transcript levels were also reflected in NIS protein abundance, which reached levels identical to the control upon AZD6244 treatment in NRAS-mutant cells but was only partially restored in cells expressing the BRAF mutant (Figure 3C). In addition, co-treatment with NSC23766 considerably impaired the AZD6244-mediated recovery of NIS protein abundance in cells expressing NRAS Q61R but did not visibly affect the extent of NIS recovery by AZD6244 in BRAF V600E-expressing cells (Figure 3C).

### 3.4. RAC1 Activity Potentiates the Positive Impact on NIS Expression Induced by MEK Inhibition in the Context of Malignant Thyroid with BRAF V600E Genetic Background

Thus far, our data suggested that the increased recovery of NIS expression induced by MEK inhibitors in mutant NRAS backgrounds, compared to those with mutant BRAF, appears to depend on a higher RAC1 activation. Thus, we hypothesized that it might be possible to improve the response of BRAF V600E-positive thyroid cancer cells to MEK inhibitors by increasing RAC1 activity. To test this, we overexpressed active-RAC1 in the PTC-derived, BRAF V600E-mutant cell line, BCPAP, and analyzed its response to MEK inhibitors, regarding NIS expression.

Consistent to that observed in PCCL3 and FRLT5 cells, a positive impact on NIS expression was observed upon treatment of BCPAP with either the MEK inhibitors AZD6244 or CH5126766. This effect was further enhanced by the co-expression of constitutively active RAC1-L61 mutant, enabling an up to six-fold increase in NIS transcript levels compared to control conditions (Figure 4).

### 3.5. RAC1 Activity Modulates the Effect of MEK Signaling on Iodide Uptake

Next, we investigated whether NIS-mediated iodide uptake is affected by the interplay between RAC1 activity and activated NRAS or BRAF signaling. For this, NRAS Q61R- and BRAF V600E-expressing PCCL3 cells, modified to stably express an halide sensitive YFP protein, whose fluorescence is quenched in the presence of intracellular iodide [32,37,38], were treated with AZD6244 and NSC23766 and analyzed by live cell fluorescence microscopy after addition of extracellular iodide (Figure 5A). Consistent with the data described above, NRAS Q61R and BRAF V600E expression led to a significant decrease in iodide uptake, which was reversed upon treatment with AZD6244 (Figure 5A,B).

Moreover, in NRAS Q61R cells, a significant reduction in NIS-mediated iodide uptake was observed upon MEK and RAC1 co-inhibition (Figure 5A,B), which was consistent with the observed decrease in NIS protein levels in the same experimental conditions (see Figure 3C). Surprisingly, the same co-treatment also reduced iodide uptake in BRAF V600E cells, where RAC1 inhibition did not affect the levels of NIS protein rescued by AZD6244 (Figure 3C and Figure 5A,B). One possible explanation for this observation would be that, besides enhancing NIS expression at transcriptional level, RAC1 signaling could be also involved in post-translational regulatory mechanisms that further promote NIS trafficking to, and function at, the plasma membrane. This prompted us to investigate whether the inhibition of RAC1 affected the abundance of NIS protein at the plasma membrane of thyroid cells. Using a cell surface protein biotinylation assay, TSH-induced NIS protein abundance at the plasma membrane of PCCL3 cells was monitored in the presence and absence of NSC23766. Consistent with the iodide influx data, the inhibition of RAC1 clearly reduced the abundance of NIS protein at the cell surface, having a much lesser impact in the overall abundance of the symporter (Figure 5C).

### 3.6. RAC1 Signaling Modulates the Impact of MEK Activation on NIS Post-Translational Events

While our findings support that MEK inhibition in association with the stimulation of RAC1 signaling can revert the downregulation of NIS functional expression in thyroid cells bearing NRAS and BRAF constitutively active mutants, it remained to be clarified to what extent these signaling cues impacted NIS post-translational regulation. Hence, to restrict our analysis to post-translational events, avoiding potential interferences resulting from NIS transcriptional regulation, we used an experimental strategy in which we evaluated the impact of MEK and RAC1 signaling on the plasma membrane abundance of exogenously expressed NIS. For that we chose the TPC1 cell line, a PTC-derived cell model, negative for both BRAF- and RAS-activating mutations, engineered to stably express a functional HA-tagged NIS protein via a CMV promoter [33]. TPC1 cells are negative for both BRAF- and RAS-activating mutations, although harboring the RET/PTC1 rearrangement that leads to a constitutively active form of the RET TK receptor. While the RET/PTC1 rearrangement can stimulate the MAPK pathway in thyroid cancer cells [10], the transduction of either NRAS Q61R or BRAF V600E into TPC1 cells induced a strong, ~three-fold increase in the endogenous level of ERK1/2 phosphorylation (Figure 6A), consistent with a robust upregulation of MAPK pathway activation and MEK activity in these cells. In addition, and similar to that observed in PCCL3 and FRLT5 cells (Figure 1B and Appendix A), the transduction of NRAS Q61R, but not of BRAF V600E, also induced and increased in endogenous RAC1 activation (Figure 6A). Notably, constitutively active NRAS increased MEK activity but also increased in nearly 2-fold HA-NIS surface levels, whereas BRAF V600E, despite also activating MEK to a similar, if no higher, degree, had no significant impact on HA-NIS abundance at the cell surface (Figure 6B). Again, since only NRAS promoted RAC1 activity, we asked whether RAC1 signaling alone would have a significant posttranslational impact on the cell surface abundance of the ectopically expressed symporter. Consistently, the inhibition of endogenous RAC1 signaling via NSC23766 treatment on parental HA-NIS TPC1 cells induced a significant decrease in HA-NIS surface levels, while these were not affected by AZD6244 treatment alone (Figure 6C). A similar effect was observed in PCCL3 cells when analyzing the impact on endogenous NIS expression of MEK inhibition alone or combined with RAC1 inhibition (Appendix A).

## 4. Discussion

With RAI being the adjuvant treatment of choice for DTC metastatic disease, a significant proportion of metastatic DTCs become refractory to RAI. This is mainly due to the defective functional expression of NIS, which, in turn, has been associated with several pathways linked to malignant transformation. Increasing NIS expression and function through the targeted inhibition of some of these pathways has, therefore, been the main objective of several preclinical and clinical studies. Since the MAPK pathway is often constitutively activated in DTC and is associated with impaired NIS expression and function, most studies have focused on inhibiting this pathway [39]. Some of these inhibitors have clearly been shown to increase NIS expression, even allowing to increase RAI uptake in particular cases. However, its effectiveness in reversing iodine refractory status proved to be clinically insufficient for all tested agents. Notably, although the MEK inhibitor AZD6244 (Selumetinib) had promising results in the clinical setting, it seemed to be most effective in tumors harboring NRAS-activating mutations rather than in those with the BRAF V600E mutation, despite both mutations being able to trigger MEK signaling. Indeed, a phase II clinical trial showed that the AZD6244-induced RAI resensitization, and its consequent therapeutic benefit, was markedly superior in patients with NRAS-mutant tumors than in those harboring the BRAF V600E-activating mutation [25]. Accordingly, in the present study, we showed that although both NRAS- and BRAF-activating mutations have a negative impact on NIS expression, NIS recovery induced by MEK inhibitors is considerably higher in NRAS Q61R- than in BRAF V600E-expressing thyroid cells, consistent with the differential drug-responses observed in patients [15].

The differential impact on NIS observed for NRAS- and BRAF-activating mutations may be justified by an additional factor promoting NIS expression that is triggered by the activation of NRAS but not of BRAF. We have previously shown that NIS expression in thyroid can be enhanced by RAC1 activity [32]. Notably, here we show that only RAS but not BRAF is able to promote the activation of RAC1, and that the blockage of RAC1 signaling is able to counteract the rescue of NIS expression induced by MEK inhibition only in the context of an NRAS activated background. In fact, while RAC1 downregulation abrogates the effect of MEK inhibition on NIS expression downstream of mutant NRAS, it produces virtually no effect when MEK is activated by BRAF V600E. These in vitro data suggest that the rescue of NIS induced by MEK inhibition in RAS-mutant cells may be potentiated by RAS-mediated activation of RAC1 (Figure 7).

In further agreement with this hypothesis is the fact that RAS-activating mutations have a less deleterious impact on NIS expression than the BRAF V600E mutation, as supported by the findings of Tavares and coworkers [15]. These authors, through the analysis of PTC data retrieved from The Cancer Genome Atlas (TCGA) database, showed that NIS expression was significantly higher in RAS-mutated PTCs compared to those carrying the BRAF V600E mutation, albeit both subgroups displayed significantly lower NIS expression compared to PTCs not harboring either RAS- or BRAF-activating mutations [15]. 

While the constitutively active BRAF V600E mutant is unaffected by the negative feedback from ERK to RAF, RAS mutants signal via nonmutated RAF proteins that respond to ERK feedback, which may attenuate the overall MAPK signaling [40]. Beyond this, the ERK negative feedback to RAF also has repercussions on the overall impact of MEK inhibitors on the MAPK pathway. Indeed, the acute inhibition of ERK activity (as it happens upon AZD6244-mediated MEK inhibition) causes a relief of its negative feedback towards RAS. This, in turn, promotes the signaling events upstream of RAF, including RAS activation, thus restarting MEK activity even in the presence of the inhibitor [41]. This effect is particularly marked when BRAF is constitutively active. CH5126766, another allosteric inhibitor of MEK, was shown to prevent BRAF reactivation by inducing MEK to adopt a conformation that impairs its release from BRAF, thereby inhibiting the BRAF kinase as well [36]. This results in the sustained inhibition of ERK signaling, which was reported to have a marked positive impact on iodide uptake in mouse models of BRAF V600E-induced TC [42]. Nevertheless, in our models, the inhibition of MEK mediated by either CH5126766 or AZD6244 led to similar results, with the greatest rescue of NIS expression being observed in the NRAS-mutated background. This, once again, supports that a positive regulatory trigger resulting from NRAS activity may be a relevant signaling cue to increase NIS expression. This also suggests that the recovery of NIS expression by MEK inhibitors in a BRAF-activated context could benefit from this positive regulation, which can be accomplished by the enhancement of RAC1 activity. Consistently, our findings revealed that in a thyroid carcinoma cell model with BRAF V600E genetic background, the upregulation of RAC1 activity is able to potentiate the positive impact on NIS expression induced by MEK inhibition.

Bringing in an additional perspective, it might be worth exploring the hypothesis of evaluating endogenous RAC1 activity as a predictor of the response to MEK inhibitors in the recovery of NIS expression, particularly in a BRAF V600E positive TC context. Jensen and co-workers, by using an anti-RAC1-GTP antibody, performed the immunohistochemical detection of RAC1-GTP in tissue sections of invasive ductal breast carcinoma from patients. The subsequent comparison between the adjacent tumor and normal tissue disclosed a correlation between RAS activation and active RAC1 levels [43]. This experimental approach could be of great value to study associations between RAC1 activation status and NIS rescue mediated by MEK inhibition in DTC. If a robust association can be established, the level of endogenous RAC1 activity in the tumor could constitute a potentially informative marker, predictive of tumor response to MEK inhibitors in the clinical setting.

Besides having a positive impact on NIS expression at the transcriptional level, our findings also support that RAC1 signaling has a role in the modulation of NIS post-translational events. In fact, the blockage of RAC1 signaling abrogated the recovery in iodide uptake observed upon MEK inhibition in the context of both NRAS-and BRAF-activating mutations. This was in contrast with what we observed for NIS expression levels, for which the impact of RAC1 inhibition was negligible in the BRAF V600E genetic background. These observations are consistent with an effect of RAC1 signaling on NIS posttranslational regulation and trafficking to the cell surface, since the inhibition of RAC1 downregulated iodide uptake without significantly affecting NIS overall levels. This was further corroborated by the analysis of these signaling events on the surface levels of the ectopically expressed NIS protein (that bypasses NIS native transcriptional regulatory events) in the PTC-derived cell line TPC1. This cell line, although negative for both BRAF and RAS activating mutations, harbors the RET/PTC1 rearrangement. This results in the production of a constitutively active chimeric RET tyrosine kinase receptor that can signal through the RAS/RAF/MEK/MAPK pathway, leading to its upregulation [10]. Notwithstanding, the overexpression in this cell line of either NRAS or BRAF constitutively active mutants was able to strongly increase the activation of MEK, indicating that the cell line was prone to further activation. Importantly, mutant NRAS, but not mutant BRAF, also significantly upregulated the abundance of ectopic NIS at the cell surface, an effect that was promptly abrogated when the mutant NRAS-induced RAC1 overactivation was suppressed with RAC1 inhibitor NSC23766. These findings strongly support that RAC1 signaling, but not MEK signaling, participates in the post-translational mechanisms that regulate NIS residence in the plasma membrane of thyroid cells. Indeed, we observed the same effect with endogenous NIS in PCCL3 cells. The abundance at the plasma membrane of TSH-stimulated NIS in these cells was drastically reduced upon RAC1 inhibition but unaffected by MEK inhibition.

Altogether, our data support that the rescue of NIS induced by MEK inhibition in RAS-mutant cells may be potentiated by the RAS-mediated activation of RAC1. Confronting this with that observed for RAI resensitization induced by MEK inhibition in RAS-mutant tumors [25], one may postulate that the RAS-mediated activation of RAC1 can positively impact the NIS functional expression at both transcriptional and post-transcriptional levels.

## 5. Conclusions

In summary, our results support that the inhibition of the MAPK pathway (which releases NIS transcription repression) when associated with an elevated RAC1 activity (which potentiates NIS expression and abundance at the plasma membrane) can promote the functional rescue of NIS, which ultimately may result in a better response to MEK inhibitors. Further studies will now be required to understand how best to translate these findings into the clinical setting.

## Figures and Tables

**Figure 1 cancers-13-05861-f001:**
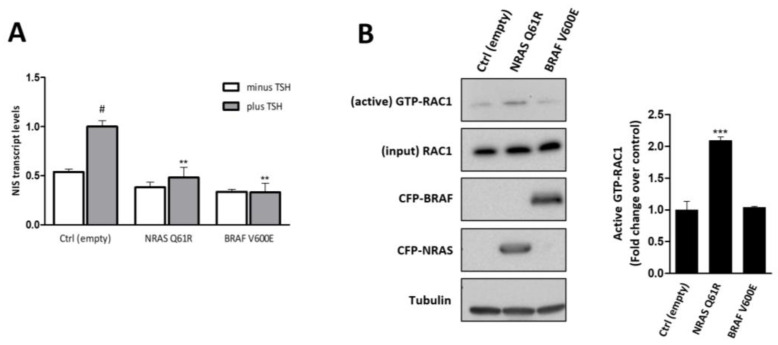
Effect of NRAS and BRAF gain-of-function mutants on NIS transcriptional expression in non-neoplastic, TSH-responsive thyroid follicular cell line. (**A**) PCCL3 cells were transduced with empty, NRAS Q61R or BRAF V600E constructs and subjected to a 24 h starvation period (minus TSH) followed by stimulation with TSH (1 mU/mL for 48 h plus TSH). NIS mRNA levels were quantified by RT-qPCR and plotted as fold differences relative to mock transfected [Ctrl (empty)] cells, treated with TSH. Values are the mean ± standard error of the mean (SEM) of five independent assays. One-way ANOVA analysis detected significant differences between the different conditions (F = 9.95; *p* < 0.001). Post-hoc Dunnett’s tests were used to identify significant variations from Ctrl (empty) conditions (^#^
*p* ≤ 0.05 relative to minus TSH; ** *p* ≤ 0.01 relative to plus TSH). (**B**) Endogenous RAC1 activation status in Ctrl (empty)-, NRAS Q61R- and BRAF V600E-expressing cells was assessed by monitoring the active, GTP-bound fraction of RAC1 by CRIB domain pull-down assay (quantified in the plot to the right; *** *p* ≤ 0.001 relative to ctrl[empty]). Total (input) and active RAC1 levels were assessed by western blot using anti-RAC1 primary antibody. NRAS and BRAF expression was detected using anti-GFP primary antibody. Alpha-tubulin was used as loading control (see Appendix A for uncropped WB).

**Figure 2 cancers-13-05861-f002:**
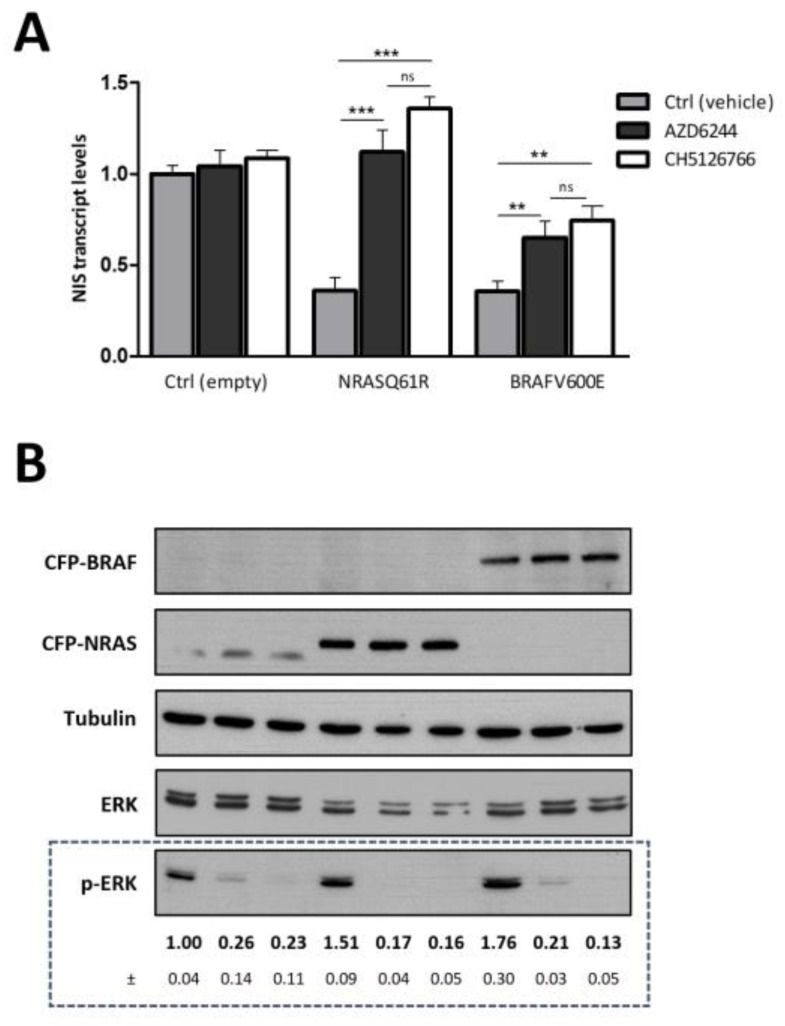
Effect of MEK1/2 inhibition on NIS transcriptional expression in thyroid cells expressing NRAS and BRAF gain-of-function mutants. (**A**) PCCL3 cells transduced as indicated were serum-starved for 24 h followed by stimulation with TSH (1 mU/mL for 48 h) in the presence or absence of either AZD6244 (10 µM for 48 h) or CH5126766 (10 µM for 48 h). NIS mRNA levels were quantified by RT-qPCR and plotted as fold differences relative to mock (CFP)-transduced cells, without treatment with the MEK inhibitors [Ctrl (empty)]. Values are means ± SEM of five independent assays. Comparisons were made using a two-tailed Student’s *t*-tests between AZD6244-treated or CH5126766-treated versus untreated setting (ns—not significant; ** *p* ≤ 0.01; *** *p* ≤ 0.001). (**B**) NRAS and BRAF expression was assessed by western blot using anti-GFP primary antibody. MAPK activation status was assessed by monitoring phosphorylated ERK 1/2 vs. total ERK 1/2 levels by western blot [quantified below the respective panel (means ± SEM of three independent assays)], using anti-phospho ERK 1/2 and anti-ERK 1/2 primary antibodies, respectively. Alpha-tubulin was used as loading control (see Appendix A for uncropped WB).

**Figure 3 cancers-13-05861-f003:**
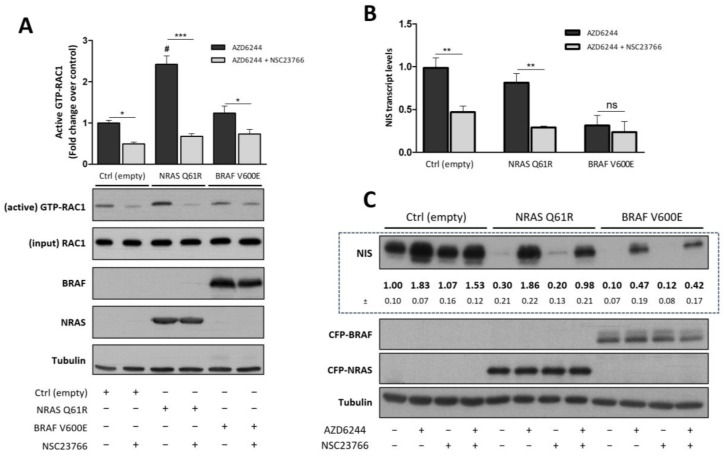
Effect of inhibiting MEK and RAC1 signaling on NIS expression. Transduced PCCL3 were serum-starved for 24 h, stimulated with TSH (1 mU/mL for 48 h; TSH plus) and treated with AZD6244 (10 µM for 48 h), either in the presence or absence of RAC1 inhibitor (NSC23766; 100 µM for 24 h). (**A**) Inhibition of endogenous RAC1 activity upon NSC23766 treatment was assessed by monitoring the active GTP-bound fraction of RAC1 by PAK-CRIB domain pull-down assay. Total (input) and active RAC1 levels were assessed by western blot using anti-RAC1 primary antibody. NRAS and BRAF expression was assessed by western blot using anti-GFP primary antibody. Alpha-tubulin was used as loading control. Graph shows means ± SEM of three independent assays. One-way ANOVA detected significant differences among the different conditions (F = 29.90, *p* < 0.0001). Tukey’s posttests were used to identify significant variations relative to mock transduced [Control (empty); ^#^
*p* ≤ 0.001] and NSC23766-treated versus untreated conditions (* *p* ≤ 0.05; *** *p* ≤ 0.001). (**B**) NIS mRNA levels were quantified by RT-qPCR and plotted as in Figure 1A. Values are means ± SEM of three independent assays. Comparisons were made using a two-tailed Student’s *t*-test between NSC23766-treated versus untreated setting (ns—not significant; ** *p* ≤ 0.01). (**C**) Endogenous NIS protein expression upon AZD6244 and NSC23766 treatments was monitored by western blot using an anti-NIS primary antibody [quantified below the respective panel (means ± SEM of three independent assays)]. NRAS and BRAF expression was assessed by western blot using anti-GFP primary antibody, and endogenous alpha-tubulin expression was used as loading control (see Appendix A for uncropped WB).

**Figure 4 cancers-13-05861-f004:**
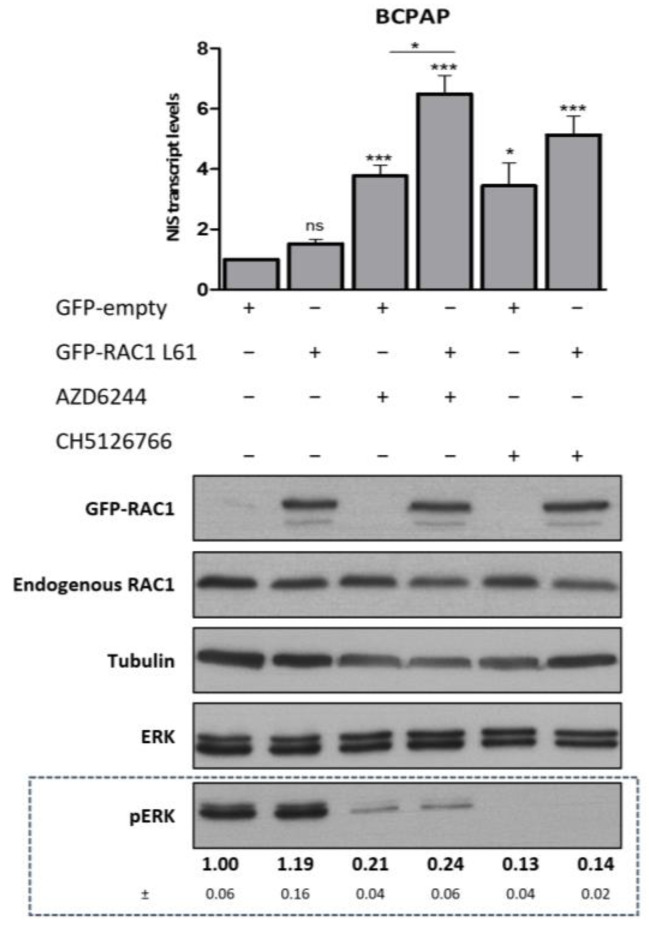
Effect of RAC1 activity on NIS transcriptional rescue by MEK inhibitors in a BRAF-mutant follicular thyroid cancer-derived cell line. The PTC-derived BCPAP cell line was transiently transfected with GFP-empty vector or GFP-RAC1 L61 expressing constructs in the presence or absence of either AZD6244 (10 μM for 48 h) or CH5126766 (10 µM for 48 h). NIS mRNA levels were quantified by RT-qPCR and plotted as fold-changes relative to the set of untreated samples transfect with GFP-empty vector. Values are means ± SEM of five independent assays. One-way ANOVA analysis detected significant differences between the different conditions (F = 14.08; *p* < 0.001). Post-hoc Tuckey’s tests were used to identify significant variations relative to control (GFP-transfected, untreated) or AZD6244 single treatment (ns—not significant; * *p* ≤ 0.05; *** *p* ≤ 0.001). Endogenous RAC1 and the ectopic expression of RAC1 L61 were monitored by western blot, using anti-RAC1 and anti-GFP primary antibodies, respectively; MAPK activation status was assessed by monitoring phosphorylated ERK 1/2 vs. total ERK 1/2 levels by western blot [quantified below the respective panel (means ± SEM of three independent assays)] using anti-phospho ERK 1/2 and anti-ERK 1/2 primary antibodies, respectively (see Appendix A for uncropped WB).

**Figure 5 cancers-13-05861-f005:**
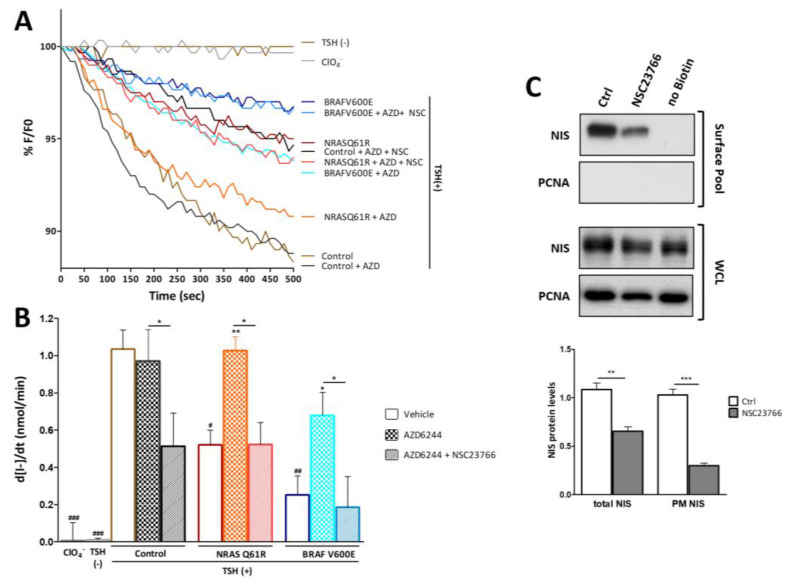
Impact of MEK and RAC1 signaling on endogenous NIS levels at the plasma membrane and NIS-mediated iodide uptake in PCCL3 cells. PCCL3 cells stably expressing the YFP-halide sensor were either mock transduced (control) or transduced with NRAS Q61R or BRAF V600E constructs, subjected to a 24 h starvation period and then treated with TSH for 96 h in the presence or absence of AZD6244 (10 µM for 48 h), alone or in combination with NSC23766 (100 µM for 24 h). YFP fluorescence was recorded continuously for 500 s, acquiring an image every 10 s, after exposure to 1 mM NaI. (**A**) Fluorescence (F) was plotted over time as percentage of fluorescence at time 0 (F_0_). Shown are representative fluorescence decay curves for each experimental condition. (**B**) Iodide influx rates calculated by fitting the curves to the exponential decay function to derive the maximal slope that corresponds to initial I^−^ influx rates. Data are means ± SEM of at least three independent assays. One-way ANOVA analysis detected significant differences between treatments (F = 8.79 and *p* < 0.0001). Post-hoc Tukey’s tests were used to identify significant variations relative to either TSH-stimulated, mock-transduced, untreated (vehicle) cells (# *p* ≤ 0.05, ^##^
*p* ≤ 0.01 and ^###^
*p* ≤ 0.001) or among equally transduced, TSH-stimulated cells treated with the indicated compounds (* *p* ≤ 0.05; ** *p* ≤ 0.01). (**C**) PCCL3 cells were treated with TSH for 48 h in the presence or absence of NSC23766 (100 µM for 24 h). Cell surface proteins were biotinylated and both whole-cell lysates (WCL) and surface protein fraction were analyzed by western blot to detect endogenous NIS protein, using α-NIS primary antibody (see Appendix A for uncropped WB). The ‘no Biotin’ condition, corresponding to cells that were not incubated with biotin, was used as control for non-biotinylated protein capture. PCNA expression served as loading (WCL) and intracellular protein contamination control (Surface pool). WB bands were quantified by densitometry analysis using Image J software (NIH, Bethesda, MD, USA). Comparisons between Ctrl or NSC23766-treated cells were made using a two-tailed Student’s *t*-test (** *p* ≤ 0.01; *** *p* ≤ 0.001).

**Figure 6 cancers-13-05861-f006:**
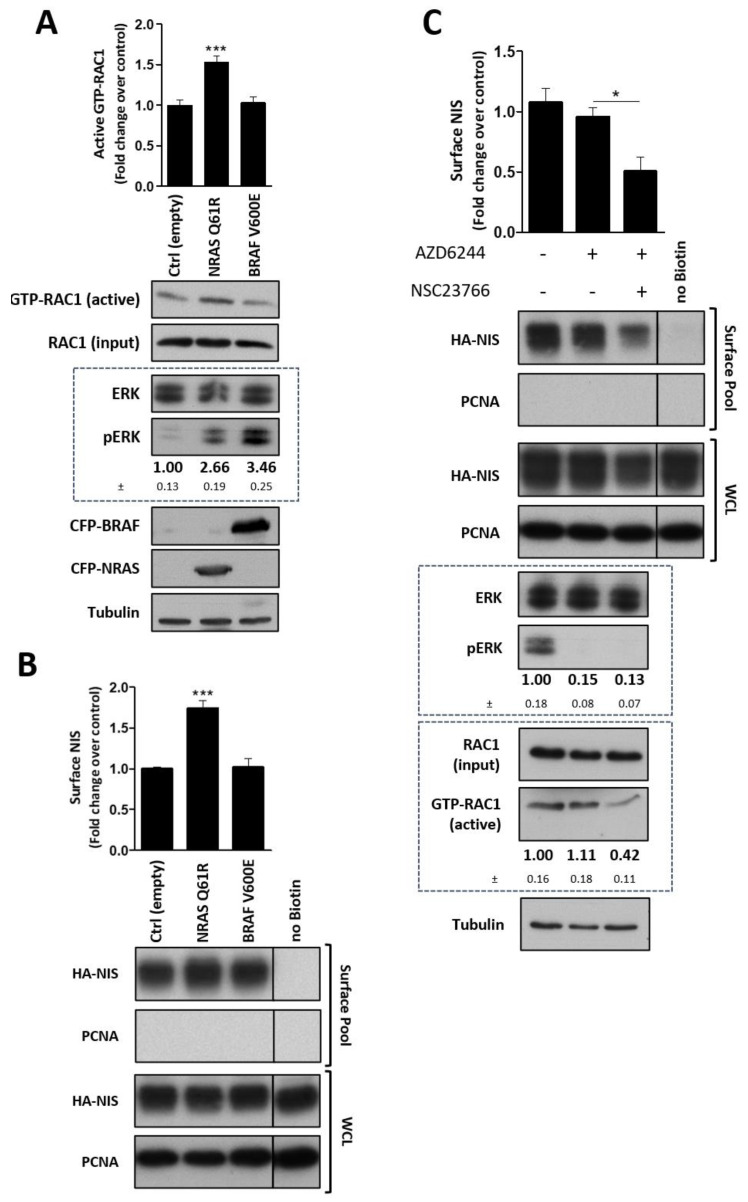
Impact of MEK and RAC1 signaling in NIS post-translational expression. The PTC-derived TPC1 cell line was modified to stably express full-length NIS construct containing a triple HA tag. (**A**,**B**) HA-NIS-TPC1 cells were transfected with CFP-empty vector [Ctrl (empty)], CFP-NRAS Q61R or CFP-BRAF V600E expressing constructs. (**A**) The activation status of endogenous RAC1 was monitored using a CRIB domain pull-down assay. Total (input) and active RAC1 levels were assessed using anti-RAC1 primary antibody. Tubulin was used as loading control. The ectopic expression of NRAS and BRAF variants was monitored in the WCL, using anti-GFP primary antibody. MAPK activation status was assessed by monitoring phosphorylated ERK 1/2 vs. total ERK 1/2 levels by western blot [quantified below the respective panel (means ± SEM of three independent assays)], using anti-phospho ERK 1/2 and anti-ERK 1/2 primary antibodies, respectively. (**B**) Cell surface proteins were biotinylated and both whole-cell lysates (WCL) and surface protein fraction were analyzed by western blot. ‘no Biotin’ condition, corresponding to cells that were not incubated with biotin, was used as control for the capture of non-biotinylated proteins. Total and surface HA-NIS protein levels were detected using anti-HA primary antibody. PCNA expression served as loading (WCL) and negative control (Surface pool). (**C**) HA-NIS-TPC1 cells were treated with AZD6244 (10 μM for 1 h) in the presence or absence of the RAC1 inhibitor, NSC23766 (100 μM for 1 h) and analyzed as in (**A**,**B**). Then, in all panels, WB bands were quantified by densitometry analysis using ImageJ software (NIH, Bethesda, MD, USA; see Appendix A for uncropped WB). Plotted values are means ± SEM of five independent assays. One-way ANOVA analysis detected significant differences between conditions [(**A**) F = 17.36, *p* < 0.001; (**B**) F = 29.50; *p* < 0.001; (**C**) F = 5.33, *p* = 0.034]. Post-hoc Tukeys’s tests were used to identify significant variations relative to control conditions (* *p* ≤ 0.05; *** *p* ≤ 0.001).

**Figure 7 cancers-13-05861-f007:**
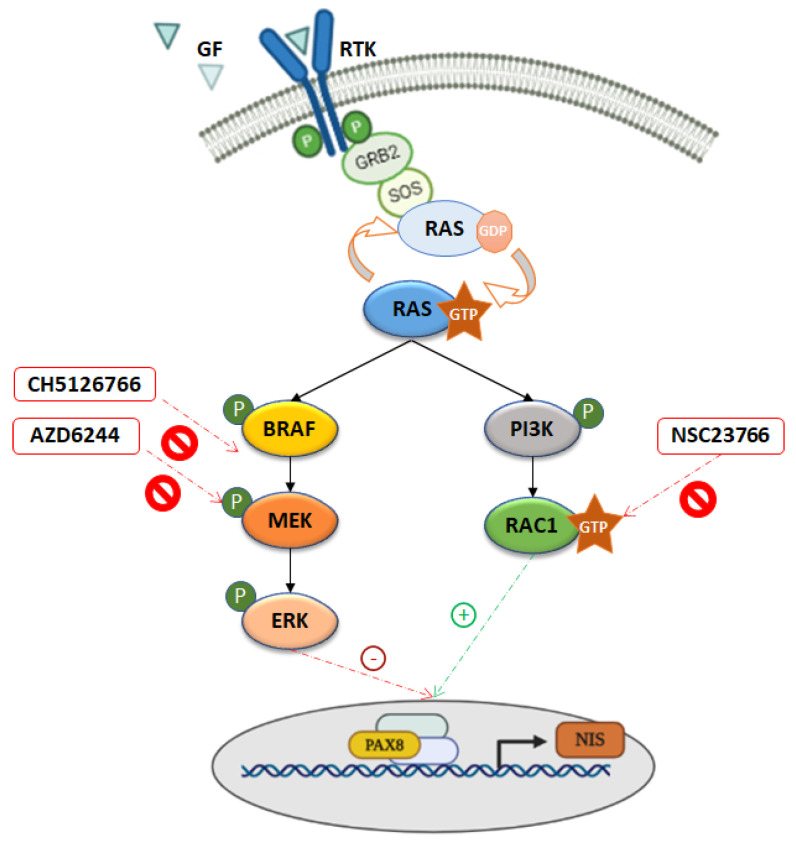
Working model for the opposite effects of MAPK and RAC1 signaling on NIS expression in thyroid cancer cells. Both NRAS- and BRAF-mutants are negative regulators of NIS expression and function; treatment with MEK1/2 inhibitor has a more meaningful positive impact in NRAS Q61R- than in BRAF V600E-expressing cells; NRAS, but not BRAF, stimulates RAC1 activity, which in turn, has a positive impact on NIS functional expression. The concerted action of MAPK pathway inhibition (which releases the repression of NIS transcription) and an increased RAC1 activity (which potentiates NIS expression and its plasma membrane abundance) could explain the better response to selumetinib treatment of thyroid tumors harboring NRAS-activating mutations over those carrying BRAF mutation (where NIS transcription is only partially rescued).

## Data Availability

The data is contained within the article or supplementary material.

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
