# Peer review of "MAPK Inhibition Requires Active RAC1 Signaling to Effectively Improve Iodide Uptake by Thyroid Follicular Cells"

_cancers, 2021, doi:10.3390/cancers13225861_

Round 1

Reviewer 1 Report

The authors addressed my comments and other reviewers comments well. which improved the manuscript. 

Reviewer 2 Report

The authors answered to all the raised points , only few WB not quantified: WB in fig 1B should be quantified

Reviewer 3 Report

This study describes the mechanism of re-expression of NIS what is really important for RAI treatment. 

For the further studies it is suggested to use two human normal thyroid follicular epithelial cell lines: Nthy-ori 3-1 and HTori-3 which are commonly available. 

Moreover, results could be validated on bigger panel of human DTC cell lines.

This manuscript is a resubmission of an earlier submission. The following is a list of the peer review reports and author responses from that submission.

Round 1

Reviewer 1 Report

 It is an interesting follow-up study of their own. Here, Faria et al reporting the role for RAC1 signaling in enhancing MAPK-inhibition to revert/ resensitize radio-iodine therapy for radioiodine refractory thyroid cancer. This study is a well-thought-out study with well-planned experiments to clearly demonstrate the role of RAC1 in NRAS-Q61R and BRAF-V600E mutation-induced thyroid cancer cells in vitro.

Minor comments

Reference 22 and 28 are the same.

Please review and update the references. Here are several references that you should consider including:

HER inhibitor promotes BRAF/MEK inhibitor-induced redifferentiation in papillary thyroid cancer harboring BRAFV600E.

Inhibitors Enhance HDAC Inhibitor-Induced Redifferentiation in Papillary Thyroid Cancer Cells Harboring BRAF (V600E): An In vitro Study

Reviewer 2 Report

In the paper " MAPK inhibition requires active RAC1 signaling to effectively 2 improve iodide uptake by thyroid follicular cells ” Márcia Faria et al., provides new information on the role of RAC1 signaling  in the rescue of NIS induced by MEK inhibition in RAS-mutant cells.  Although the study are performed only in vitro, authors  performed a good experimental design to demonstrated the role of RAC1 signaling in enhancing  MAPK-inhibition in the context of RAI therapy in DTC and the main conclusions are properly supported by the data.  The manuscript is well written and clear to the readers. However, the following minor suggestions are recommended to improve the quality of the manuscript:

General comment: the authors did not follow the instructions from the journals: materials and methods section should be moved after discussion section then also references should be corrected accordingly.  

  1. Page 3, “Cell lines and culture conditions”: please add the HEK293T (human embryonic kidney) cell line  used for lentiviral assay.  
  1. Page 5, results: in my opinion in the results part of a manuscript only the results should be written. Instead, in this manuscript, authors described results and gave comments. For example in lines 283-291. They should used them in the discussion part.

Reviewer 3 Report

IS is crucial iodide transporter, allowing the transport of radioactive iodine (RAI) in differentiated thyroid carcinoma (DTC) treatment. In this manuscript, the authors showed that inhibition of RAC1 abrogates the effect of MAPK inhibition on NIS expression. Therefore, the authors asserted that the RAC1 has an important role in enhancing MAPK inhibition, thereby modulating RAC1 signaling could be a new opportunity for therapeutic intervention in DTC. Overall, the authors should conduct much more experiments to support the idea of this study, and the data presentation is not suitable for publication. This manuscript is not recommended for publication in this journal.   

Comments to the authors:

  1. The authors only used rat thyroid cells (PCCL3 and FRTL5). The authors should check the role of RAC1 using human thyroid cancer cells.
  2. This study lacks in vivo The authors should examine if modulating RAC1 signaling enhances the effect of MAPK inhibition and check the NIS expression in the mouse model.
  3. It would be better to check the NIS protein level in Figs.1A, 2A, 3B, and 4.
  4. The authors should quantify the WB results.

Reviewer 4 Report

In the manuscript “MAPK inhibition requires active RAC1 signaling to effectively improve iodide uptake by thyroid follicular cells” Márcia Faria and co-workers analyses the effect of MAPK inhibition on NIS expression and activity in two non-neoplastic thyroid cell lines transduced with NRAS or BRAF oncogene, as well as the effect of RAC inhibition on the rescue of NIS expression mediated by the same MEK inhibitors.

Moreover, they show that the expression of constitutively active RAC enhances the effect of MEK inhibitors on increasing NIS expression by using the thyroid tumor cell line BCPAP, characterized by BRAF oncogene. Then, to study a possible impact of RAC on NIS post-translational events, they transfect SLC5A5 gene encoding NIS in thyroid cells to show that inhibition of RAC signaling induces a significant reduction of membrane NIS level.

The experimental design is generally clear. The Results are interesting also because of the important role of NIS in thyroid cancer treatment.

The experiments are generally well conducted, but some points must be addressed.

The major point of weakness is in the final experiments: the Authors, to study the post translational effects of RAC on NIS, use a tumor cell line, TPC1 (expressing exogenous NIS) and transduce NRAS and BRAF oncogenes similarly to what performed in PCCL3 and FRTL5 cells, derived from normal epithelium. Instead TPC1, that indeed as claimed by the Authors is negative for both NRAS and BRAF oncogenes, express RET/PTC1 oncogene, a constitutively active form of RET TK receptor, able to activate a neoplastic signaling, including RAS/ERK and PI3K/AKT pathways. To avoid interference by RETPTC1 signaling on NRAS, BRAF and RAC effects on NIS, the same experiments should be done in a non-neoplastic cell line, like rat PCCL3 or FRTL5, or human NThy.

Other important points:

-The WB analysis in Figure 1B is not quantified: the active GTP-RAC1 must be quantified in NRAS vs control cells.

-The quantification of results is necessary for all the shown Western Blots

-In Figure 5 the treatments of control cells are lacking. This do not allow to understand whether the effect of RAC inhibitor on NIS activity is specific for transduced cells or not.

-The Discussion should be shortened, limiting speculations, and discussion in Results section must be also reduced/limited: e. g. lanes 283-294

Minor points

-In Figure1A the NIS expression in the oncogene transduced cells is not analysed without TSH. This do not allow to understand if TSHR is still active.

-Due to the opposite effect of RAC1b respect to RAC1 on NIS expression (published by the same Authors) it would be interesting to know its expression level in BCPAP cells.